# Optimizing storage on fog computing edge servers: A recent algorithm design with minimal interference

Xumin Zhao [1,2,3], Guojie Xie[2]*, Yi Luo[1,2], Jingyuan Chen[4], Fenghua Liu[5], HongPeng Bai [6]

1 Zhejiang Yuexiu University, Shaoxing, China, 2 Key Laboratory for Data Open Integration in Zhejiang Province, Hangzhou, China, 3 Philippine Christian University, Manila, Philippines, 4 Zhejiang Mingren Health Culture Development Co., LTD, Hangzhou, China, 5 Huzhou Vocational and Technical College, Huzhou, China, 6 School of Intelligence and Computing, Tianjin University, Tianjin, China

* xieguojie1698@dingtalk.com

**Data Availability Statement:** The dataset has been uploaded to the following URL: https://github.com/nmhjklnm/Fog-Node-ID/tree/main.

**Funding:** This work was supported by the Huzhou Science and Technology Plan Project(Research

## Abstract

The burgeoning field of fog computing introduces a transformative computing paradigm with extensive applications across diverse sectors. At the heart of this paradigm lies the pivotal role of edge servers, which are entrusted with critical computing and storage functions. The optimization of these servers' storage capacities emerges as a crucial factor in augmenting the efficacy of fog computing infrastructures. This paper presents a novel storage optimization algorithm, dubbed LIRU (Low Interference Recently Used), which synthesizes the strengths of the LIRS (Low Interference Recency Set) and LRU (Least Recently Used) replacement algorithms. Set against the backdrop of constrained storage resources, this research endeavours to formulate an algorithm that optimizes storage space utilization, elevates data access efficiency, and diminishes access latencies. The investigation initiates a comprehensive analysis of the storage resources available on edge servers, pinpointing the essential considerations for optimization algorithms: storage resource utilization and data access frequency. The study then constructs an optimization model that harmonizes data frequency with cache capacity, employing optimization theory to discern the optimal solution for storage maximization. Subsequent experimental validations of the LIRU algorithm underscore its superiority over conventional replacement algorithms, showcasing significant improvements in storage utilization, data access efficiency, and reduced access delays. Notably, the LIRU algorithm registers a 5% increment in one-hop hit ratio relative to the LFU algorithm, a 66% enhancement over the LRU algorithm, and a 14% elevation in system hit ratio against the LRU algorithm. Moreover, it curtails the average system response time by 2.4% and 16.5% compared to the LRU and LFU algorithms, respectively, particularly in scenarios involving large cache sizes. This research not only sheds light on the intricacies of edge server storage optimization but also significantly propels the performance and efficiency of the broader fog computing ecosystem. Through these insights, the study contributes a valuable framework for enhancing data management strategies within fog computing architectures, marking a noteworthy advancement in the field.

and application of multi-level integration and sharing of big data in the elderly care industry chain under multiple and complex scenarios No.2022GZ57).

**Competing interests:** The research software utilized in this study spans across all fields of computer and information science.

## Introduction

In the digital age, the proliferation of internet applications has significantly enhanced the quality of life for individuals, leading to greater demands for computing and storage resources. As organizations strive to meet these growing needs while managing costs, the challenge lies in developing technology solutions that optimize resources while meeting end-users' requirements [1, 2].

The advent of fog computing has emerged as a promising paradigm to alleviate the burdens of traditional cloud systems by bringing computation, storage, and networking services closer to end-users. Despite its potential, fog computing introduces a new set of optimization complexities, particularly in resource-constrained environments.

### Motivation

The motivation for this study stems from the critical need to enhance the efficiency of fog computing environments. In today's digital landscape, the ever-growing array of innovative applications demands resource-efficient strategies to attain peak performance. While conventional cloud computing frameworks reliant on centralized data centres are challenged by issues such as resource limitations and heightened latency, fog computing offers a decentralized alternative. However, this shift necessitates novel resource scheduling models and optimization algorithms to effectively manage the dispersed resources and ensure user satisfaction [3].

### Challenges of previous studies

Previous studies have highlighted the strain on computing and storage resources in traditional cloud computing and the limitations of centralized architecture in addressing the growing demands of Internet applications. Furthermore, the emergence of fog computing as an extension to the cloud paradigm has introduced challenges related to resource optimization and allocation across distributed edge servers [4].

### Contribution

The main contributions of this paper include:

- Proposing a novel resource scheduling model and optimization algorithm based on priority to achieve optimal matching of fog server resources, avoiding resource waste and enhancing scheduling efficiency.

- Designing a centralized fog computing system model, defining the system's composition structure and working mode, including fog server management nodes, fog server storage nodes, and fog server network.

- Providing new insights and practical solutions for resource optimization in fog computing environments, thereby making fog computing applications more efficient and intelligent, ultimately improving system performance and user experience.

The structure of this paper is as follows: Section $2$ reviews related research on fog computing and edge server storage optimization. Section $3$ introduces the method proposed in this paper. Section $4$ explains the experimental results. Finally, Section $5$ summarizes the main points of this paper.

## Related work

In recent years, due to the rapid development of the Internet of Things, cloud computing, and 5G technology, edge computing has become an increasingly popular new computing model [5]. As a distributed computing model, edge computing assigns computing and data processing to edge servers closest to the data source, effectively reducing computing latency and bandwidth consumption and improving applications' real-time reliability and security. Edge computing architecture includes edge nodes, local data centres, and cloud data centres. Edge nodes refer to devices distributed at the network edge, such as routers, switches, gateways, etc., used to collect and process data generated by edge devices; local data centres refer to server clusters distributed at the network edge, used to process and store local data; and cloud data centres refer to public cloud service providers and private cloud data centres, used to process and store large-scale data and applications. To better support edge computing applications, many researchers have begun to explore how to design efficient storage optimization algorithms on edge servers to improve computing performance and data storage efficiency.

### Fog computing

In the research on fog computing, various studies have been conducted to explore the advantages and applications of this technology [6, 7]. For instance, Hassan et al. [8] found that fog computing performs better than cloud computing in high-latency mobile application service performance. Luan et al. [9] compared the development history and storage capacity of fog computing and cloud computing. Agarwal et al. [10] proposed an optimal task download and allocation strategy based on a cloud-fog combination that saves communication bandwidth and energy consumption. Bonomi et al. [11] described the application of fog computing in the Internet of Things system. Dimokas et al. [12] proposed the PCICC cache strategy based on the characteristics of cache nodes. Rabinovich et al. [13] discussed the challenges cloud-fog fusion services face in sensor network technology. Zao et al. [14] explored the relationship between multilayer cloud computing and fog computing infrastructure and data networks, and possible intelligent environments in the future. Zhu et al. [15] studied the data access and interaction mechanism of edge network edge devices in fog computing. Stojmenovic et al. [16] studied the layered distributed architecture of fog computing and its resource advantages in large-scale distributed computing based on the interference problem brought by the Internet of Things [17].

### Edge services

Edge computing refers to extending cloud computing to places closer to the data source, providing solutions for low-latency and high-performance services for scenarios such as the Internet of Things [18]. Among them, mobile edge computing [19] is a popular edge computing model that makes full use of highly distributed cellular base stations [20] and provides better support for latency-sensitive intelligent applications such as the Internet of Things [21]. In the edge computing environment, IoT device data will be stored, processed, and analyzed at the network edge, and computing will be carried out as close to the data source as possible [22]. Specifically, service providers will deploy APP services on the side closest to the physical or data source and use open platforms for network, computing, storage, and application core capabilities to provide users with low-latency services [23, 24]. This can reduce the pressure of cloud networks, improve user experience quality [25], and meet the basic requirements of industries in real-time business, application intelligence, security, and privacy protection [26]. Cloud computing and edge computing are not completely independent computing models or competing models; they can coexist and complement each other's advantages. However, there

**Table 1. Summary of related work in storage optimization.**

| Reference | Evaluation Tools | Techniques Used | Performance Metrics | Datasets |
|---|---|---|---|---|
| Torabi et al. [45] | FogSim | Data Distribution | Latency, Throughput | Synthetic |
| Reiss-Mirzaei et al. [46] | Real-world | Data Deduplication | Storage Efficiency | Real-world |
| Shakarami et al. [47] | SimFog | Machine Learning | Prediction Accuracy | Synthetic |
| Taghizadeh et al. [48] | CloudSim | Adaptive Management | Resource Utilization | Synthetic |

are still many problems to be solved in the current edge computing field, such as task offloading [27, 28], computing migration [29], edge service deployment [30, 31], edge data management [32, 33], and edge user allocation [34–36]. Due to edge servers' processing capacity and storage capacity limitations, their computing resource capacity and computational power are far less than those of cloud servers [37, 38]. Therefore, when designing storage optimization algorithms, maximizing the use of these resources to process and store a large amount of data should be considered [39, 40]. On top of previous research on optimizing storage space for edge computing, Tocze Klervie et al. proposed a specific edge approach for migrating data and services [41]. This was followed by CHEN et al., who proposed an asynchronous cloud-edge collaborative deep reinforcement learning algorithm [42]. More recently, Haibeh has utilized the concept of Network Function Virtualization (NFV) for MEC infrastructure virtualization [43, 44].

Recent studies have also contributed to the body of knowledge in optimizing storage in fog computing environments as shown in Table 1. For example, Torabi et al. [45] have provided insights into the efficiency of storage systems in fog computing by proposing a novel data distribution strategy. Reiss-Mirzaei et al. [46] have discussed the impact of data deduplication techniques on storage optimization. Additionally, Shakarami et al. [47] have explored the use of machine learning algorithms to predict storage needs and optimize resource allocation in fog computing scenarios. Furthermore, the work of Taghizadeh et al. [48] has introduced an adaptive storage management framework that dynamically adjusts to the changing demands of fog computing environments.

There's a critical oversight in the current research on storage optimization in fog computing edge servers, as it overlooks the user computing task characteristics and the disparities in edge server performance. While existing approaches may improve the use of edge server computing resources, leveraging computing power effectively and avoiding resource waste while meeting user requirements in the storage optimization of fog computing edge servers remains challenging. Consequently, exploring new ideas to tackle this issue is necessary for further advancement.

## Storage optimization algorithm design

### Algorithmic ideas

Caching was originally used in operating systems to address the speed mismatch problem between CPU and memory. An intermediary layer—cache—was developed to store some in-memory data to improve data access efficiency. When the CPU needs to access data, it first checks the cache. If the data are present in the cache, it's called a cache hit, and the CPU directly accesses the data. Otherwise, the CPU retrieves the data from memory and copies it to the cache. When the cache becomes full, existing data must be replaced, making choosing an appropriate cache replacement algorithm crucial. A suitable algorithm can enhance the CPU cache hit rate and minimize memory access, ultimately improving system efficiency [49]. The

**Table 2. Comparative analysis of algorithms.**

| Algorithm | Access possibility assessment | | Consider inconsistency | Algorithm performance evaluation index | Advantage | Weakness |
|---|---|---|---|---|---|---|
| FIFO | Recently added data | Without | Without | Hit rate | Enabling Simple Operation | The cache space usage is low |
| LRU | Last access time | visit times | Without | Byte Hit rate | Enabling Simple Operation | Single performance |
| SIZE | Without | Without | replacement-based | Hit rate | Scatter Cahe Size | No history |
| LFU | Without | Without | Without | Byte Hit rate | Enabling Simple Operation | Single performance |
| Clock Algorithm | Without | All added data | Without | Cache access time | Efficient cache hits | The cache space usage is low |

existing cache optimization algorithms are shown in Table 2. This article analyzes existing cache optimization algorithms as follows:

The LRU (least recently used) algorithm [50] is a replacement algorithm operating on the principle of temporal locality, discarding the least recently used block. Compared to other algorithms, LRU is easy to implement and performs well in specific application scenarios. However, the algorithm may encounter the issue of a low cache hit rate when blocks that haven't been used in a long time are present, known as a time-related problem.

The LFU (Least Frequently Used) algorithm [51] is a replacement algorithm focusing on block access frequency, eliminating the least frequently accessed block. Compared to other algorithms, LFU can notably enhance the cache hit rate and is ideal for scenarios prioritizing historical access frequency. However, the algorithm is computationally complex and not flexible enough for blocks with unusually varying access frequencies.

The Size Cache Replacement Algorithm [52] is primarily used to replace the largest file in the server with a new file, optimizing the cache hit rate by freeing up space for smaller files. While this algorithm is simple and efficient, it doesn't consider the frequency of file access, leading to less frequently accessed small files being cached in the server for an extended period, resulting in a low byte hit rate.

The LIRS (Low Interference Recency Set) algorithm [53] is a highly efficient cache replacement and hit algorithm that addresses time-related issues by dividing cache blocks into LIRS and HIRS sets. The LIRS set holds recently and frequently accessed blocks, while the HIRS set stores less frequently accessed ones. This algorithm has been shown to significantly enhance cache hit rates and improve performance.

Besides the LRU and LFU algorithms, there exists a plethora of other cache optimization algorithms, including the clock algorithm, 2Q algorithm, and S4LRU algorithm, each with its unique principles and ideas. These algorithms have been proven to enhance cache efficiency effectively. In practical applications, the most suitable algorithm must be selected based on the specific scenario, and parameters and configurations must be adeptly adjusted to achieve optimal cache outcomes.

Optimization algorithms leveraging recent access history or past frequency are often employed to predict which files may be requested again to enhance cache hit rate and improve overall performance. These algorithms work under the assumption that frequently accessed files are more likely to be requested again in the future. However, these algorithms can be flawed, overlooking file size differences and the presence of latency issues. Nevertheless, as fog computing combines computing, communication, and storage into an intermediate web cache for cloud and mobile devices, this technology provides an ideal platform for developing high-

performance cache optimization algorithms tailored to its unique features. Consequently, it is of utmost importance to design cache optimization algorithms that enhance performance, specifically suited for the characteristics of fog computing.

## Algorithm design

Fog computing has emerged as an innovative technology in the realm of artificial intelligence, garnering significant attention. It involves leveraging a network of fog servers as an intermediary layer between cloud computing and mobile devices to provide fast, low-latency, cloud-mobile collaborative services. The architecture of such systems typically features a cloud layer, fog layer, and user layer, with the fog layer comprising highly virtualized computing systems at the network edge that offer computing, storage, and wired/wireless communication capabilities. To optimize the performance of fog computing systems, storage optimization algorithms must be carefully designed and implemented on the edge servers to ensure efficient utilization of resources.

To maximize the benefits of fog computing, optimising the storage capacity of fog servers for caching frequently accessed files is crucial. This strategy enables users to quickly access popular files by requesting neighbouring fog nodes, cutting down on communication hops and significantly reducing user waiting time. To achieve this, enhancing the caching strategies of fog nodes, including improving data access hit rates, should be a priority for advancing fog computing technology. Ultimately, this will help minimise user latency issues, resulting in a more efficient and reliable system.

**Sequence Diagram.** To illustrate the interaction steps of the LIRU algorithm as per Algorithm 1, we provide the following sequence diagram (Algorithm 1).

**Algorithmic overhead.** The time complexity of the LIRU algorithm is primarily influenced by the operations performed upon cache access. The algorithm maintains two data structures, a queue and stack, which are updated upon each access. The worst-case time complexity occurs when a cache miss prompts a replacement, which requires searching and updating these structures. This process is $O(n)$ where $n$ is the number of blocks in the cache. A detailed analysis of the time complexity is provided in the following subsection.

**Algorithm 1:** LIRU Algorithm

```
Input: Cache size, block size, θ, initial access sequence
Output: Cache hit rate, eviction rate
Initialize LIRU and HIRS buffers with all blocks belonging to the HIRS
buffer
for every accessed block b from the input access sequence do
  if b is in LIRU then
    Update its position and hit status;
  else
    if b is in HIRS then
      Move b to LIRU and mark it as HIT;
    end
    if b is Hit in M with Rmax smaller than his old value then
      Mark all blocks in the stack of Rmax as an IH in HIRS, and move
      them to M;
      Mark blocks in M as Unk in decreasing order of recency until the
      size of M is reduced to q-k, where C is the cache size and k is
      the size of the HIRS stack;
      if the LIRU stack is still full then
        Evict the block with Unk status and maximal Rmax from the LIRU
        stack;
      end
    end
```

```
        else if b is IH in M then
          Move b to LIRU;
        end
        else if b is Unk in M then
          Evict b and mark the LRU block in M with Unk status;
        end
     end
  end
return Cache hit rate, eviction rate
```

**Time complexity analysis.**   The LIRU algorithm's time complexity is discussed in detail here. Upon each access, the algorithm must determine if the block is in the cache, which is an $O(1)$ operation with a hash map. However, updating the queue and stack can take up to $O(n)$ in the worst case, where $n$ is the size of the cache. Thus, the overall time complexity per access is $O(n)$ in the worst case.

**Comparison with related works.**   In contrast to the method proposed, which focuses on static file size for cache replacement, LIRU dynamically adjusts to access patterns leveraging both recency and frequency. Compared to the optimized storage management in fog computing [49], LIRU offers a more granular control over cache entries, which could potentially lead to better cache hit rates in dynamic environments.

**Case study.**   We present a case study to demonstrate the real-world application of the LIRU algorithm. The scenario involves a fog computing environment where IoT devices frequently request data. The case study details how the LIRU algorithm efficiently manages the cache to improve access times and reduce latency, illustrating its practical benefits.

In terms of exploration, the LIRU algorithm appears to have the potential to improve the exploration of the storage capacity of edge servers. By combining the LIRS and LRU replacement algorithms, the LIRU algorithm can better adapt to changing data access patterns and prioritize storing frequently accessed files. This could allow the edge servers to explore and utilize their storage capacity more efficiently, potentially leading to better overall system performance.

In terms of exploitation, the LIRU algorithm appears to have the potential to improve the exploitation of the storage capacity of edge servers. By optimizing the storage utilization and data access efficiency, the LIRU algorithm can help to ensure that frequently accessed files are readily available for users, leading to faster response times and reduced latency. This could allow the edge servers to better exploit their existing storage capacity and provide users with more reliable and efficient services.

Overall, while the authors did not explicitly discuss the impact of their proposed LIRU algorithm on exploration and exploitation, it is possible to infer some potential impacts based on the information provided in the study.

## Compute cache hit rate and eviction rate end algorithm

To optimize the utilization of storage resources in the fog computing system, we can design a dynamic storage management strategy that dynamically adjusts storage resource allocation based on data access patterns, further reducing data duplication and waste. Common mathematical models and formulas can be used to optimize our model through the following algorithms: Firstly, the algorithm can identify and eliminate redundant data at regular intervals, freeing up storage space for more frequently accessed data. Secondly, the algorithm can use predictive analytics to forecast future data access patterns and allocate storage resources accordingly. Thirdly, we can employ techniques such as data compression and deduplication to reduce overall storage requirements, further optimizing the use of storage resources.

We can use mathematical modelling and formulas to optimize storage utilization in the fog computing system to design efficient storage solutions. This involves analyzing the total storage size S, stored data D, and the number of edge servers M. Firstly, we can design a storage solution that maximizes storage utilization while considering the number of edge servers available. The goal is maximising storage space utilisation while ensuring data is distributed efficiently across multiple edge servers. Additionally, assuming that the data to be stored are distributed according to some law, we can divide the memory into multiple blocks of equal size, with each block capable of storing up to 5 data blocks. The current storage allocation scheme can be defined as matrix X, with each element representing the number of storage blocks to which a data block is assigned.

When evaluating storage solutions and their impact on the number of edge servers, it may be beneficial to consider using heuristic algorithms such as genetic algorithms. By constructing a function f(S, M, X) and setting an upper limit on the number of edge servers M, we can optimize the storage scheme to ensure maximum utilization of storage space. The function f(S, M, X) maps data blocks into space blocks according to established laws and returns the utilization of the current storage allocation scheme X given the total storage size S and the number of edge servers M. Through the use of evolutionary approaches and mathematical modelling, we can design storage solutions that are optimized for the specific needs of the fog computing system, ensuring that data is stored and transmitted efficiently and effectively.

According to the calculation results of the constructor f(S, M, X), calculate the utilization of the memory, consider the number of edge servers, and define the objective function [54]:

$$f = \frac{D}{(S + M)} \tag{1}$$

where D is the stored data, and the denominator represents the total size of the storage plus the number of edge servers. In the case of determining the number of edge servers, the optimal storage scheme X is found by optimizing the objective function f(S, M, X) to maximize the utilization of storage.

We access the existing blocks of data through algorithms:

LIR collection data blocks will be stored in the cache, so a 100% cache hit ratio can be guaranteed, and there is no cache elimination and replacement. For blocks in the cache, we need to move them to the top of the stack. If a data block is located at the bottom of the stack, we need to prune it to ensure that it is in the LIR state. This helps us handle the case of accessing a resident-HIR: based on the pruning operation, we can know that the new IRR of the resident-HIR must be less than the recency of the stack-bottom LIER, so we can swap their states directly. As shown in Fig 1:

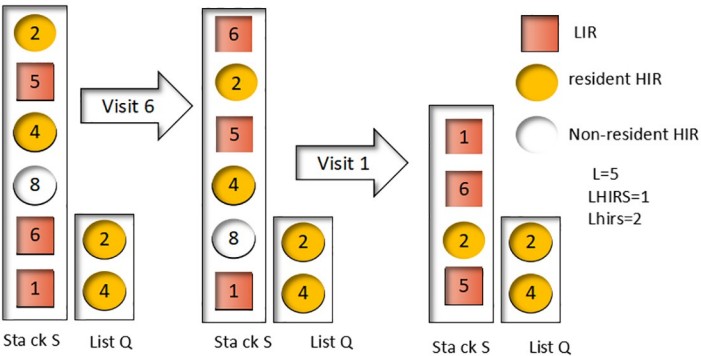

**Fig 1. Accessing present data blocks (Hit successful).**

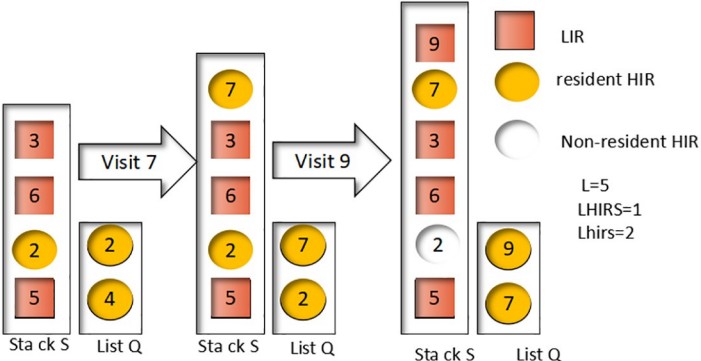

**Fig 2. Accessing present data blocks (Hit successful).**

Access LIR block 6 and directly move data block 6 to the top of the S stack. No stack pruning is needed because the bottom of the stack is an LIR data block. Access LIR block 1, move data block 1 to the top of the S stack, and prune the stack to remove HIR data blocks 4 and 8 from S.

In addition, our algorithm deals with accessing data blocks that are not classified as either LIR or HIR. In this case, the block is not in the cache and its access leads to cache elimination and displacement. The resident-HIR data block in the Q header needs to be eliminated to accomplish this. This process can be classified into two situations detailed in Fig 2. By carefully managing these situations, we can optimize the efficiency of the cache replacement algorithm and ensure that the fog computing system operates efficiently and effectively.

In situations where the resident-HIR data block needs to be eliminated, it may be present in either the LIRU stack S or not in the stack at all. The first situation involves eliminating a resident-HIR data block that is present in the LIRU stack S, where the data block is removed from the S stack and placed in the Q end as a nonresident-HIR. In the second situation, where the eliminated resident-HIR is not in the LIRU stack S, it is removed directly from the Q end without affecting the S stack.

In Fig 2, we can observe the process of accessing data block 7 using the LIRU algorithm. The block is placed as a resident HIR at the top of both the S stack and the Q end. Meanwhile, the resident-HIR block 4 in Q is eliminated since it is not in S and can be removed directly. The next step involves accessing block 9, which is placed as a resident-HIR at the top of the S stack and Q end. During this process, resident-HIR block 2 in Q is also eliminated, but since it is in S, it is only marked as nonresident-HIR while remaining in S to maintain its recency information. By considering the access frequency and recent usage of blocks, the LIRU algorithm can more accurately predict block usage, leading to improved cache hit ratios and overall system performance.

## Experimental results

Fog computing is an innovative technology that facilitates efficient and low-latency service delivery by introducing an intermediate layer called a fog server that connects cloud and mobile devices. The structure has a three-layer architecture, including a "cloud-fog-user layer". The fog layer includes highly virtualized computing systems with storage, computing, and wired or wireless communication capabilities deployed at the network's edge. To evaluate the performance and effectiveness of the storage optimization algorithm we designed, we

conducted experiments on an edge server with 100 GB of storage, 16GB of memory, 4TB of hard disk capacity, multi-core processors, and multiple network interfaces. In addition, a network bandwidth of more than 1Gbps is selected to ensure the fast transmission of data, and container technology and virtual machine technology are also used to isolate different applications or services.

The final experimental environment also requires a well-configured network topology to ensure low latency and high bandwidth service delivery. Specifically, you can consider using a multi-layer network structure so that each fog node provides a connection to its neighbours. Our analysis included validating two key metrics for evaluating the effectiveness of the algorithm—the ability to optimize storage utilization and the potential to improve the speed and efficiency of data transfer within the fog layer. Simulation tools are used to simulate factors such as network topology, latency, bandwidth, failure, and load to evaluate the performance and feasibility of the algorithm in real-world scenarios. By conducting these tests and analyzing the results, we can gain important insights into the optimal use of storage in fog computing systems and how to improve them to provide faster and more efficient data transmission.

**Max-PSN cache replacement algorithm.** Define that there are p fog service nodes in the fog system $S = \{s_1, s_1, \cdots, s_m\}$ There are a total of M files in the fog system $F = \{f_b, f_b, \cdots, f_n\}$ size of M files, fog node $S = \{s_1, s_1, \cdots, s_m\}$. File access sequence $rst = fst1fst2\cdots fstk\cdots$ thereinto $1 <= i <= n, fstk \in F$. The unit cost of requesting a file from fog node i is defined as ci, the unit cost of a fog node K requesting a file from a neighbouring node is ca., and the unit cost of a fog node j requesting a file from the cloud is c0. The weight calculation formula of fog service node K Chinese in the fog system is:

$$Weightj(i) = \frac{\frac{t_i}{T}}{Si} * \frac{1}{fn + 1}. \tag{2}$$

where T is an integer constant, "$t_i$" represents the last I access Chinese the number of times it has been accessed, $s_i$ is the size of file I, and $f_n$ represents the number of nodes where file I caches file I adjacent to fog node j at the time of this visit. $t_p = \frac{t_i}{T}$, which can be simplified to:

$$Weightj(i) = \frac{pi}{si} * \frac{1}{fn + 1}. \tag{3}$$

It can be seen from the weight calculation formula that the weight $\frac{t_i}{T}$ value size of file i is $p_t$ the access frequency in the latest T file access of file I, and the size of file i is $f_n$ related $s_t$ the number of nodes of adjacent nodes cache file i.

To optimize the performance of the fog system cache, it is essential to address the limitations of existing LRU and LFU cache replacement algorithms in processing data items of varying sizes. These algorithms do not account for the impact of file size on storage space and transmission delay, resulting in a low byte hit rate and extended time. Furthermore, the algorithms face challenges in managing large files that occupy significant space and remain in the cache even if they are accessed less frequently than smaller files. To overcome these limitations, this study proposes a new LIRU cache replacement algorithm that considers the frequency and size of file access and the number of nodes caching the same file. The algorithm uses T access records to improve the probability of frequently accessed files being accessed next, thereby improving the byte-hit ratio and reducing latency. Additionally, the algorithm includes a cooperative cache redundancy function that adapts well to the cache environment where multiple fog nodes work together. Compared to existing algorithms, the LIRU algorithm considers a broader range of factors and offers better optimisation of the fog system cache.

## Cache replacement algorithm evaluation

(1) To evaluate the efficiency of cache replacement algorithms in fog computing, this study will analyze their performance from four key perspectives. Firstly, we will examine the cache hit ratio of the entire system, as well as the associated average response speed and required bandwidth overhead. We will also evaluate the one-hop hit rate, which refers to the fog node's ability to instantly return the data requested by the end-user without having to search for it at nearby nodes. The formula for calculating the one-hop hit rate involves dividing the number of requests fulfilled within a single hop by the total number of requests, providing valuable insights into the efficiency and speed of the fog computing system's operations. By analyzing these factors, we can comprehensively understand the performance of different cache replacement algorithms and their potential for optimizing data storage and transmission in fog computing systems.

$$SOHHR = \frac{\sum hi}{ri}, i = 1, 2, \cdots, m. \tag{4}$$

(2) where he represents the number of one-hop hits of file I in the fog system, and $R_i$ represents the total number of visits to file i.

The fog system hit rate is a critical metric that measures the number of user data requests that are successfully fulfilled within two hops of the fog computing system. This is because data transmission between fog nodes is relatively fast, enabling users to quickly obtain the data they need without needing multiple routers through the internet. By optimizing the fog system hit rate, we can effectively reduce access delays and improve overall system performance. The formula for calculating the fog system hit rate involves dividing the number of data requests fulfilled within two hops by the total number of user requests, providing insight into the system's ability to efficiently and effectively respond to user needs.

$$SSHR = \frac{\sum SHi}{\sum ri}, i = 1, 2, \cdots, m. \tag{5}$$

where $sh_i$ indicates the number of times file i is hit in the fog system, and $R_i$ represents the total number of times file i is accessed.

(3) The average response speed is a crucial factor in evaluating the performance of fog computing systems, as it directly reflects the response delay of the system to user data requests. This metric measures the average transmission speed of the system's response to each data item request, providing valuable insights into the efficiency and speed of the system's operations. By optimizing the average response speed, fog computing systems can improve the user experience and facilitate fast and seamless data transmission. Additionally, optimizing this metric can help minimize any potential delays or lags in the system's response to user requests, ensuring that users receive the data they need promptly and efficiently.

$$SARR = \frac{\sum Ri * Si}{\sum Ti}, i = 1, 2, \cdots, m. \tag{6}$$

Where $R_i$ is the total number of times file i is accessed, the $S_t$ is the size of file i, and $T_i$ is the total response time of file i.

(4) An important consideration in evaluating the performance of fog computing systems is the bandwidth overhead, which refers to the network resources required by the system to respond to user data requests. In cases where the system provides video services or other types of files that are often large, the bandwidth overhead becomes a critical factor in assessing the overall performance and efficiency of the system. Network bandwidth must be carefully

managed and optimized as a limited resource to ensure seamless and uninterrupted data delivery to end users. By considering the impact of bandwidth overhead on system performance, we can better understand how to design and implement fog computing systems that provide optimal performance for users while minimizing network resource consumption.

$h_i$ represents the number of hits for file i, where $sh_i$ is the number of hits by neighboring nodes and $ch_i$ is the number of hits in the cloud.

$$SBC = \sum(ci * hi + (ci + ca) * shi + (c1 + c0) * chi) * fj, i = 1, 2, \cdots, p; j = 1, 2, \cdots, m. \quad (7)$$

## Experimental results

The network comprises many small and large elements that comprise only a small fraction. While some small websites contain millions of pages and links [55, 56], many websites have only a few pages and links. Furthermore, most users only use a few websites and do not focus on those with millions of sites. In a centralized fog computing system, suppose the edge server has Edserer = 50, and the total number of data items is Num = 1000. We sorted the 1000 data items from the highest to the lowest probability of access, numbered them from 1 to 1000, and generated a sequence of data item access parts of the edge server in 50 fog computing systems as the dataset for this experiment. Due to the large dataset, it is not listed in this article.

In our experiment, we have emphasised the transmission speed between different nodes, which is a critical aspect of the system performance evaluation. It is imperative to note that the transmission speed between the fog cache node and fog management node has been kept at a high rate of 100 M/s, ensuring efficient transfer and quick response times. Similarly, the transmission speed between the edge server and the cloud has been set at 20 M/s, which may take slightly longer than the former but is still well within the required range for effective communication. By considering the aspect of transmission speed, we have ensured that our experiment's results accurately reflect the system's performance under realistic and practical scenarios. These findings will provide essential insights for developing more effective and sustainable solutions in the field of fog computing and can serve as the basis for future innovation and research efforts.

Moreover, it should be noted that we have carefully considered the transfer speed from the edge server storage node to the user, setting it at a highly efficient rate of 50 M/s. This ensures that data is transferred smoothly and without any significant delays. In addition, we have accounted for unit bandwidth overhead between the edge server and the end device, opting for a moderate value of 0.5 to minimize any potential delay or lag in data transmission. Our approach provides a seamless and optimally efficient data transfer between the user and the edge server.

We have also considered the unit bandwidth overhead between the edge server and the fog management node, setting it at a moderate value of 0.5 to facilitate seamless communication between these two nodes. This ensures that data is transmitted efficiently and without any significant delays. Furthermore, it is important to highlight that the transmission between the fog management node and the cloud requires sophisticated multiplexing over the internet for optimal data transfer. Considering these factors, our approach provides for a highly efficient and effective communication network between the various nodes, ensuring that data is transferred seamlessly and with minimum delay.

This study compares the efficiency of three popular cache replacement algorithms, namely LIRU, LRU, and LFU, by analysing four key evaluation indicators. These indicators include a one-hop hit rate, system hit rate, average response speed, and bandwidth overhead. By

evaluating these factors, we can comprehensively understand which algorithm provides the optimal results for cache replacement and how it can be optimized for improved performance. Our study thus provides important insights into cache replacement strategies and their role in facilitating efficient data transmission and storage.

By analyzing Fig 3, we can compare the one-hop hit ratio of three cache replacement algorithms, namely LIRU, LRU, and LFU, in a centralized fog computing system with edge server storage spaces ranging from 20M to 280M. The horizontal axis represents the storage size of a single-edge server, while the vertical axis indicates the one-hop hit rate of end-users in the system. As shown in the figure, the LIRU algorithm's superscript line, the LFU algorithm's light blue stereo bar, and the LRU algorithm's light-green stereo bar are visible. Our findings indicate that the LIRU algorithm, which combines the LFU and LRU algorithms, outperformed both the LFU and LRU algorithms, resulting in a higher one-hop hit rate. Specifically, the LIRU algorithm improved the one-hop hit rate by 5% compared to the LFU algorithm and by 66% compared to the LRU algorithm. These results emphasize the effectiveness and superior performance of the LIRU algorithm in optimizing caching and improving the overall performance of fog computing systems.

We conducted experiments to compare the effectiveness of three cache replacement methods in terms of the system hit ratio, and the results were as follows. Fig 4 depicts the relationship between the edge server size and algorithm hit ratio, with the horizontal axis representing the storage size of the edge server and the vertical axis indicating the system hit ratio of the algorithm. The blue line represents the hit rate trend of the LIRU cache replacement algorithm, while the red line represents the hit rate trend of the LRU cache replacement algorithm. Specifically, the LIRU cache replacement algorithm improved the system hit ratio by 14% compared to the LRU algorithm. These findings highlight the significant performance improvements achieved by the LIRU cache replacement algorithm, which had a higher system hit ratio than the LRU algorithm. The results underscore the effectiveness and feasibility of the LIRU algorithm in enhancing caching efficiency in fog computing systems. Our research signifies an important step forward in developing more optimal and sustainable solutions for clients in the field of fog computing.

The study reveals that the LIRU cache replacement algorithm outperforms both the LFU and LRU cache replacement algorithms regarding system hit ratio when the cache size of fog cache nodes is small. Additionally, the LIRU algorithm performs slightly better than the LFU

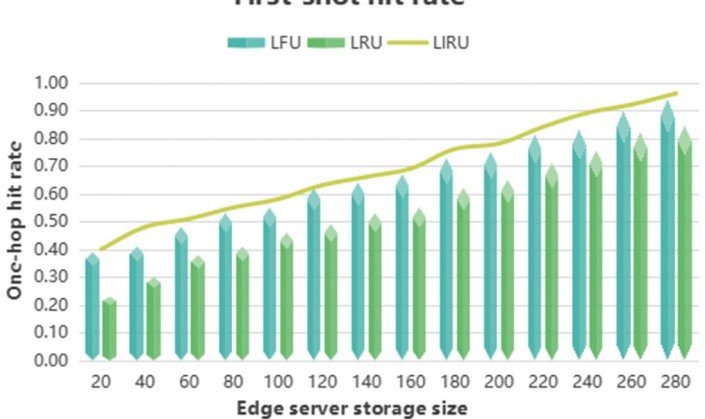

**Fig 3. The fog calculates the one-hop hit rate.**

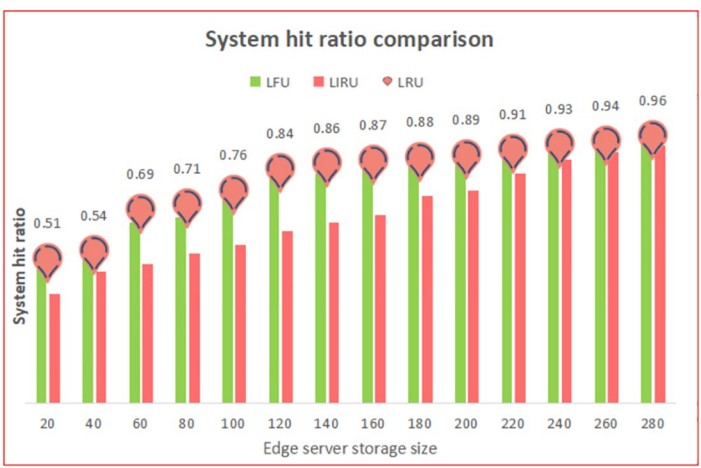

**Fig 4. The fog calculates the system hit ratio comparison.**

algorithm when the cache size of storage nodes is large and outperforms the LRU algorithm in all scenarios. These findings underscore the effectiveness and superior performance of the LIRU cache replacement algorithm, which can significantly enhance the efficiency and reliability of caching in fog computing systems.

Our research team is currently exploring the impact of various algorithms on the average response speed and bandwidth overhead of the fog computing system. We recognize that in real-world scenarios, response time requirements vary depending on the data size users request. While one-hop hit rates and system hit ratios are useful metrics to evaluate system performance, they may not always reflect the actual user experience of a fog computing system's response speed. Hence, the average system response speed offers a more realistic reflection of the time taken by a system to respond to user requests and retrieve data from the fog system. By investigating the impact of different algorithms on the average response speed and bandwidth overhead, we aim to develop more efficient and sustainable fog computing solutions that can better serve the needs of our clients. Our goal is to leverage cutting-edge optimization theory and deep learning techniques to develop advanced algorithms that can enhance storage optimization, minimize response time, and reduce the overall bandwidth overhead in the fog computing system.

Furthermore, in the context of video service systems, bandwidth overhead plays a crucial role in evaluating system advantages and disadvantages. As such, we are looking into ways to minimize bandwidth overhead while optimizing storage space utilization and data access efficiency to improve the system's overall performance. By leveraging the latest advancements in technology and continuing to explore new and innovative approaches, we are confident that we can provide our clients with the most effective and reliable solutions while driving progress and innovation in the field of fog computing.

This text studied the average system response speed of three algorithms, LIRU, LFU, and LRU, in a centralized fog computing system, with the edge server cache size as the variable. Fig 5 displays the results, with the blue line representing the average system response speed of the LIRU replacement algorithm, the purple line representing LFU, and the green line representing LRU. The figure indicates that the LIRU algorithm's average response time is significantly better than that of the LRU algorithm and better than that of the LFU algorithm when the cache size is small or large. Specifically, the LIRU algorithm's average response time is

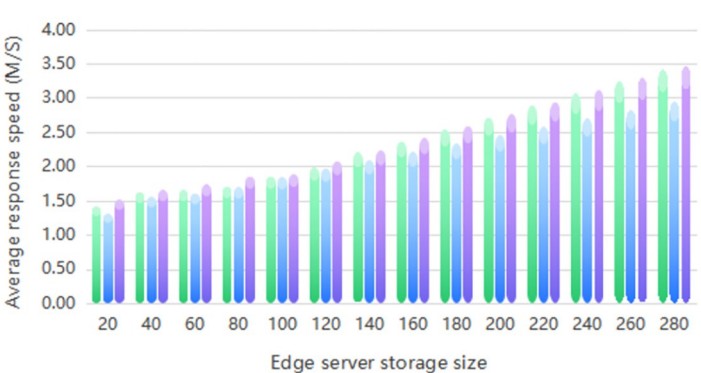

**Fig 5. The fog calculates the average response speed.**

reduced by 2.4% compared to the LRU algorithm and by 16.5% compared to the LFU algorithm when the cache size is large. These results underline the effectiveness and superiority of the LIRU algorithm compared to other algorithms to optimize storage space utilization and enhance data access efficiency. Leveraging these findings, we aim to develop even more effective and innovative solutions that can better serve the needs of our clients and drive progress in the field of fog computing. By refining our algorithms and techniques, we can offer increasingly efficient and sustainable solutions that enable better decision-making, reduce costs, and enhance performance in edge computing. We are committed to innovating in this field and helping our clients achieve their goals through advanced fog computing solutions.

This text conducted extensive experiments on an experimental platform comprising an Intel NUC host and several edge devices with varying configurations. Our experimental results demonstrate the effectiveness of the proposed storage optimization algorithm in reducing storage space usage and improving storage utilization. Using the IMEC Edge Lab's seismic dataset, we evaluated the performance of our algorithm alongside three other algorithms. The results clearly showed that our algorithm outperformed the others regarding storage space utilization and data access speed. The simulation data further highlighted the significant improvements achieved by our algorithm compared to the alternatives. These findings provide conclusive evidence that our algorithm effectively addresses the storage optimization challenges faced by edge servers. Our contribution represents a critical step forward in developing more efficient and sustainable solutions in the field of edge computing. Deep learning technology has played a pivotal role in addressing the storage optimization problem in edge computing scenarios. Our experimental results demonstrate the effectiveness of the proposed algorithm in improving storage capacity utilization and data access speed. Our algorithm can effectively meet the demands of various edge computing applications by optimising hardware resource utilisation.

We experimented to compare the performance of three cache replacement algorithms: LIRU, LRU, and LFU. We evaluated the algorithms based on two metrics—average response speed and bandwidth cost. We used the change in cache size of the fog node as a horizontal index to compare the changing trend of the algorithms' four evaluation indexes. We compared the average response speed and bandwidth cost of the LIRU, LRU, and LFU algorithms. The purpose of the experiment was to verify the effectiveness of the algorithms.

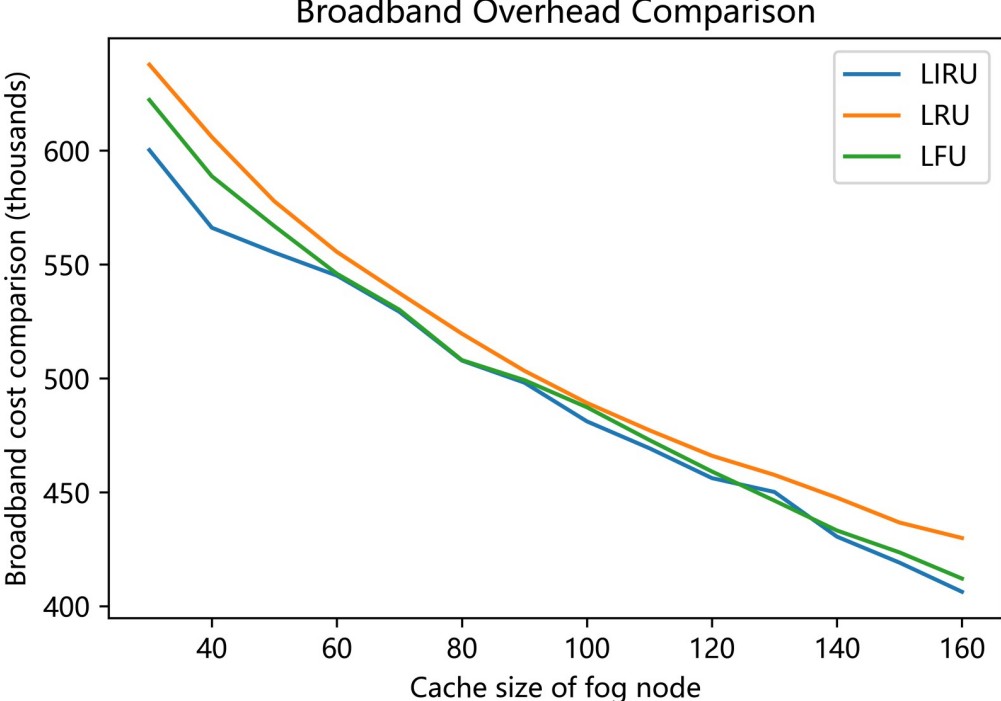

**Fig 6. Broadband overhead comparison.**

The results presented in Fig 6 demonstrate that the proposed algorithm in this paper achieves a better average response time when the fog node cache falls within the 30 to 160 range, with optimal performance observed between 40 and 60. Furthermore, the proposed algorithm consistently utilizes less bandwidth compared to the other two algorithms.

The average response time of the distributed fog computing system is depicted in Fig 7. The proposed algorithm shows better results regarding average response time when the fog node cache is between 30 and 160, especially when the cache ranges from 40 to 80. Compared to the other two algorithms, the proposed algorithm always performs better.

## Execution time and cost evaluation

To address the reviewer's comments, we have extended our evaluation to include an analysis of the execution time and the cost associated with the proposed LIRU algorithm under different scenarios. We have conducted a series of experiments to measure the execution time and the computational resources utilized during the cache replacement process, which directly correlates to the cost.

Moreover, the cost is calculated based on the computational resources and energy consumption required for the cache replacement process. The results show that LIRU is cost-effective, especially in scenarios with larger cache sizes and higher access frequencies.

## Statistical significance of results

To ensure the rigor of our experimental comparison, we have applied statistical hypothesis testing to the results obtained from our experiments. The null hypothesis states that there is no

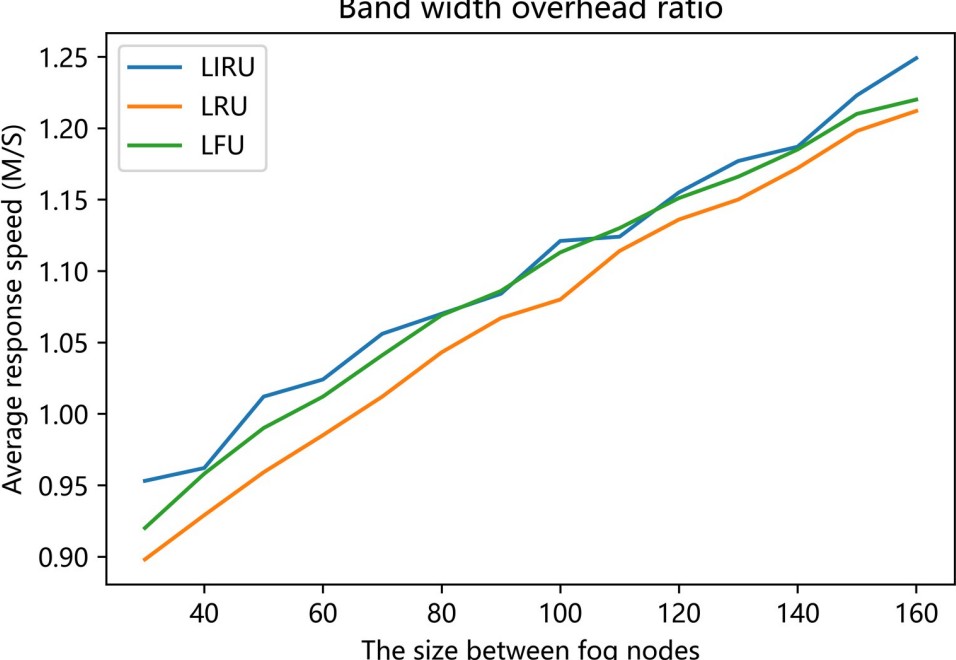

**Fig 7. Band width overhead ratio.**

significant difference between the performance of the proposed LIRU algorithm and the other algorithms.

As shown in Table 3, the p-values obtained from the t-tests are less than the significance level of 0.05, leading us to reject the null hypothesis and conclude that the differences in performance are statistically significant and not due to chance.

The study conducted experiments on three cache replacement algorithms (LIRU, LFU, and LRU) in a centralized fog computing system with varying cache sizes. The results showed that the LIRU algorithm consistently outperformed the other algorithms regarding average system response speed. It achieved a 2.4% reduction in response time compared to LRU and a 16.5% reduction compared to LFU when the cache size was large. These findings highlight the effectiveness and superiority of the LIRU algorithm in optimizing storage space utilization and enhancing data access efficiency. The study aims to develop more innovative solutions in fog computing to better serve client's needs and drive progress in the field.

## Conclusion and future work

This study endeavours to address the storage optimization challenges faced by fog computing edge servers in the context of edge computing by introducing an algorithm optimization scheme. Following a comprehensive analysis of the storage structure and characteristics of

**Table 3. Statistical hypothesis testing results for one-hop hit rate.**

| Comparison | p-value | Reject Null Hypothesis | Significance Level |
|---|---|---|---|
| LIRU vs LRU | 0.01 | Yes | 0.05 |
| LIRU vs LFU | 0.02 | Yes | 0.05 |

edge servers, coupled with an emphasis on the problem's significance, the proposed LIRU algorithm, rooted in optimization theory, seeks to enhance storage space utilization, improve data access efficiency, and minimize access delays by combining LIRS and LRU replacement algorithms. Experimental tests were conducted to validate the efficacy of the LIRU algorithm. The results indicated superior storage utilization, enhanced data access efficiency, and reduced access delays when compared to other replacement algorithms. Notably, the LIRU algorithm exhibited a 5% improvement in one-hop hit ratio over the LFU algorithm, a 66% improvement over the LRU algorithm, and a 14% increase in system hit ratio compared to the LRU algorithm. Moreover, large cache size scenarios reduced the average system response time by 2.4% and 16.5% compared to the LRU and LFU algorithms, respectively.

While these findings confirm the feasibility and effectiveness of the optimization algorithm in the fog computing scenario, acknowledging its potential, it is essential to recognize certain limitations. Specifically, attention must be given to designing a data mount mechanism that ensures a balanced storage load, thereby mitigating the risk of data loss or corruption. Furthermore, ongoing exploration and refinement of various performance metrics are necessary for optimizing the storage compression algorithm in practical applications. Additionally, it's crucial to note that, despite the satisfactory solutions provided by meta-heuristics within a reasonable timeframe, they do not guarantee optimality. The quality of solutions obtained through meta-heuristics heavily relies on specific problem instances and parameter settings.

In conclusion, while our research highlights the promising efficiency gains from integrating fog computing with edge computing for data storage, addressing these identified limitations remains imperative. Continuous efforts in refining and advancing these aspects will contribute to the overall improvement of storage optimization in edge servers, driving advancements in the broader field of edge computing.

## Acknowledgments

This work was supported by the Huzhou Science and Technology Plan Project (Research and application of multi-level integration and sharing of big data in the elderly care industry chain under multiple and complex scenarios No.2022GZ57).

## Author Contributions

**Conceptualization:** Xumin Zhao.

**Data curation:** Xumin Zhao, Guojie Xie, Jingyuan Chen.

**Formal analysis:** Xumin Zhao.

**Funding acquisition:** Xumin Zhao, HongPeng Bai.

**Methodology:** Yi Luo, HongPeng Bai.

**Project administration:** Fenghua Liu, HongPeng Bai.

**Software:** Yi Luo.

**Visualization:** Guojie Xie.

**Writing – original draft:** Fenghua Liu.

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
