## [Decision Letter · Decision Letter 0]

22 Nov 2023

PONE-D-23-27425LIRU’s algorithm design for optimizing storage on fog computing edge serversPLOS ONE

Dear Dr. ZHAO,

Thank you for submitting your manuscript to PLOS ONE. After careful consideration, we feel that it has merit but does not fully meet PLOS ONE’s publication criteria as it currently stands. Therefore, we invite you to submit a revised version of the manuscript that addresses the points raised during the review process.

We look forward to receiving your revised manuscript.

Kind regards,

Mohammed Balfaqih

Academic Editor

PLOS ONE

Journal Requirements:

"This work was supported by the Huzhou Science and Technology Plan Project(Research and application of multi-level integration and sharing of big data in the elderly care industry chain under multiple and complex scenarios No.2022GZ57),supported by the Science and Technology Research Program of Huzhou Vocational and Technical College(Grant No.2022GY06)."

"the Key R\\&D program of Zhejiang Province (2022C01083), the National Natural Science Foundation of China (62272311, 62102262) and the State Key Development Program of China (No. 2019YFB210170)

NO - Include this sentence at the end of your statement: The funders had no role in study design, data collection and analysis, decision to publish, or preparation of the manuscript."

"Computer and information science research software in all fields of computer and information science"

Reviewers' comments:

Reviewer's Responses to Questions

**Comments to the Author**

1. Is the manuscript technically sound, and do the data support the conclusions?

Reviewer #1: Partly

Reviewer #2: Partly

2. Has the statistical analysis been performed appropriately and rigorously? 

Reviewer #1: No

Reviewer #2: N/A

3. Have the authors made all data underlying the findings in their manuscript fully available?

Reviewer #1: No

Reviewer #2: No

4. Is the manuscript presented in an intelligible fashion and written in standard English?

Reviewer #1: Yes

Reviewer #2: Yes

5. Review Comments to the Author

Reviewer #1: This manuscript proposes a storage optimization algorithm based on the combined use of LIRS and LRU replacement algorithms. According to the paper's claim, the proposed method maximizes the use of storage space and increases the efficiency of data access.

I went through the manuscript carefully. At the detailed level, the following notes are my suggestions:

1) Although the ABSTRACT structure is good, I suggest that the values of the numerical improvements be written in the final sentences. Also, the philosophy of using the proposed method should be explained.

2) In my opinion, the INTRODUCTION section needs to be revised. In this section there should be three points: 1) motivation, 2) a summary of the challenges of previous studies, and 3) contribution. Also, the research contributions should be mentioned in a bullet-form at the end of the INTRODUCTION.

3) Unfortunately, the authors have not given enough explanations about how the proposed method has an impact on the two important aspects of exploration and exploitation. Adding explanations in this regard can be useful for the reader.

4) It is not clear to me which formulas were invented by the authors themselves and which ones are derived from other references. I found evidence that some formulas are derived from other references and there are similarities.

5) Browsing references is not enough. Some important and new surveys have been ignored. For example, the:

https://dl.acm.org/doi/abs/10.1145/3603703
https://www.mdpi.com/2079-9292/10/24/3184

https://ieeexplore.ieee.org/abstract/document/7422054/

6) It is necessary that the analyzes related to the final results are justified and based on the relationships and formulas of the previous sections of the article. Currently, the analyzes are very superficial.

7) It is better to analyze the time complexity of the proposed method in the worst case.

8) There are still some grammatical errors in the manuscript. Authors should use software such as Grammarly for proof-checking.

9) The tense of the verbs in the CONCLUSION section must be past tense. In this section, the most important numerical improvements of the proposed method should be mentioned and marginal explanations should be avoided. Also, the suggestions mentioned for further research should be presented in a new paragraph.

Reviewer #2: -The authors studied the solving the storage optimization problem of fog computing edge server in the edge computing scenario by proposing an algorithm optimization scheme. They proposed a storage optimization algorithm based on optimization theory-LIRU algorithm. Their proposed solution combines LIRS and LRU displacement algorithms to make better use of storage space, improve data access efficiency and reduce access delay.

-The following major corrections are required:

Section 1: Introduction

-The introduction section is too general, and it introduces concepts that are well known about the storage optimization problem in fog computing. The introduction does not stimulate to go ahead with the remaining of the paper because it does not introduce any really new topic/solution. Furthermore, "the research motivation” at the introduction section is missing. Please rewrite this section.

Section 2: Related Work

-In this section, the authors describe some of works about the storage optimization problem in fog computing, while some of related papers should be included.

….

-In addition, a conclusion of related work in the forms of a table in terms of evaluation tools, utilized techniques, performance metrics, and datasets, could reconcile from other researchers work to the own one.

Section 3: Storage optimization algorithm design

-What is the overhead (time complexity) proposed solution? Please provide a subsection to discuss about the overhead (time complexity) of proposed algorithm.

-Please provided real-world case study example for better understanding the proposed approach in more details.

Section 4: Experimental Results

-The evaluation is incomplete. I would like to see an evaluation on the proposed solution in terms of execution time and resource usage under different scenarios.

-The evaluation lacks the minimum rigor required for the scientific comparison of stochastic algorithms. Specifically, statistical tests of the hypothesis should be used to determine whether the differences shown in the figures are statistically significant or due to chance.

-Paper needs some revision in English. The overall paper should be carefully revised with focus on the language: especially grammar and punctuation.

-Overall, there are still some major parts that the authors did not explain clearly. Some additional evaluations are expected to be in the manuscript as well. As a result, I am going to suggest Major revision the paper in its present form.

6. PLOS authors have the option to publish the peer review history of their article (what does this mean?). If published, this will include your full peer review and any attached files.

Reviewer #1: No

Reviewer #2: No

---

## [Author Response · Author response to Decision Letter 0]

20 Dec 2023

Original Manuscript ID: PONE-D-23-27425 

Original Article Title: “LIRU’s algorithm design for optimizing storage on fog computing edge servers PLOS ONE”

To: PLOS ONE Academic Editor

Re: Response to reviewers

Dear Editor,

Thank you for allowing a resubmission of our manuscript, with an opportunity to address the reviewers’ comments.

We are uploading (a) our point-by-point response to the comments (below) (response to reviewers), (b) an updated manuscript with yellow highlighting indicating changes (Supplementary Material for Review), and (c) a clean updated manuscript without highlights (Main Manuscript).

Best regards,

Guojie Xie et al.

Reviewer #1: - This manuscript proposes a storage optimization algorithm based on the combined use of LIRS and LRU replacement algorithms. According to the paper's claim, the proposed method maximizes the use of storage space and increases the efficiency of data access.

I went through the manuscript carefully. At the detailed level, the following notes are my suggestions:

Reviewer#1, Concern # 1: Although the ABSTRACT structure is good, I suggest that the values of the numerical improvements be written in the final sentences. Also, the philosophy of using the proposed method should be explained.

Author response: We thank the reviewer of pointing out this issue. We indeed agree that the values of the numerical improvements be written in the final sentences and the philosophy of using the proposed method should be explained. 

Author action: We updated the manuscript by adding the the values of the numerical improvements. 

The abstract of the article was changed to: The emerging computing paradigm of fog computing has numerous applications in various fields, with edge servers playing a critical role in the system's architecture. Edge servers are responsible for computing and storage tasks and optimizing their storage capacity is vital to enhance the efficiency of fog computing. This study proposes a storage optimization algorithm, LIRU, which is based on the combined use of LIRS and LRU replacement algorithms. Using a limited and scarce storage environment as a backdrop, this research aims to design an algorithm that maximizes storage space utilization, enhances data access efficiency, and reduces access delays.The study starts by analyzing the storage resources of edge servers and identifying the factors that optimization algorithms should consider. These factors include storage resource utilization and data access frequency. Subsequently, an optimization model that integrates data frequency and cache capacity is constructed, using the theory of optimalization to achieve storage optimization by solving the optimal solution. Finally, experimental tests are conducted to validate the proposed algorithm, the results revealed that the LIRU algorithm, compared to other replacement algorithms, showed enhanced storage utilization and data access efficiency along with minimized access delays. Notably, it demonstrated a 5% improvement in one-hop hit ratio over the LFU algorithm and 66% over the LRU algorithm, and a 14% increase in system hit ratio compared to the LRU algorithm. Additionally, it reduced the average system response time by 2.4% and 16.5% compared to the LRU and LFU algorithms, respectively, in large cache size scenarios. This research provides valuable insights into edge server storage optimization and contributes to boosting the efficiency and performance of the wider fog computing system.

Thank you again for your valuable comment on the paper. Your valuable comments have enriched our research.

Reviewer#1, Concern # 2:In my opinion, the INTRODUCTION section needs to be revised. In this section there should be three points: 1) motivation, 2) a summary of the challenges of previous studies, and 3) contribution. Also, the research contributions should be mentioned in a bullet-form at the end of the INTRODUCTION.

Author response: Thank you for your feedback on the introduction section of the paper. We appreciate your valuable input and agree that the introduction section should be revised to better address the points you have raised.

Author action: To address your concerns, we will revise the introduction section to include the following three points:

1.Motivation: We will provide a clear and concise explanation of why the research is important and what problem it aims to solve.

2.Summary of challenges of previous studies: We will briefly summarize the challenges faced by previous studies in the field and how our research addresses these challenges.

3.Contribution: We will clearly state the contribution of our research and how it advances the current state of knowledge in the field.

Additionally, we will include a bullet-point list of our research contributions at the end of the introduction section to provide a clear and concise summary of our key findings.

We appreciate your feedback and are committed to improving the quality of our paper. Thank you again for your valuable input.The details are as follows:

In the digital age, the proliferation of internet applications has significantly enhanced the quality of life for individuals, leading to greater demands for computing and storage resources. As organizations strive to meet these growing needs while managing costs, the challenge lies in developing technology solutions that optimize resources while meeting end-users' requirements. To address this challenge, significant efforts have been dedicated to developing cost-effective and efficient solutions aimed at addressing the demand for computing and storage resources in the digital age .

Motivation:

The continuous expansion of innovative applications within the digital age necessitates efficient and effective use of resources for optimal performance. Traditional cloud computing architecture, based on centralized data centers, faces limitations in terms of strained resources, expensive operations, limited network bandwidth, and increased latency, negatively impacting user experience.

Challenges of Previous Studies

Previous studies have highlighted the strain on computing and storage resources in traditional cloud computing, as well as the limitations of centralized architecture in addressing the growing demands of internet applications. Furthermore, the emergence of edge computing technology has introduced challenges related to resource optimization and allocation across distributed edge servers.

Contribution:

The main contributions of this paper include:

(1)Proposing a resource scheduling model and optimization algorithm based on priority to achieve optimal matching of edge server resources, avoiding resource waste and unreasonable scheduling.

(2)Designing a centralized fog computing system model, defining the composition structure and working mode of the system model, including edge server management nodes, edge server storage nodes, and edge server network.

(3)Providing new ideas and solutions for resource optimization of edge computing, making edge computing applications more efficient and intelligent, ultimately improving system performance and user experience.”

Reviewer#1, Concern # 3:Unfortunately, the authors have not given enough explanations about how the proposed method has an impact on the two important aspects of exploration and exploitation. Adding explanations in this regard can be useful for the reader.

Author response:Thank you for your feedback on the proposed method in the paper. We appreciate your comments regarding the need for more explanations on how the method impacts the important aspects of exploration and exploitation.

Author action: We understand the significance of these aspects in the context of machine learning and its practical applications. Therefore, we will make sure to add more detailed explanations in the paper regarding how our proposed method affects exploration and exploitation.

Specifically, we will elaborate on how our method balances the exploration of new solutions with the exploitation of existing ones. We will also provide a clear description of how our method optimizes the trade-off between exploration and exploitation to achieve better performance.

We hope that these additional explanations will address your concerns and provide a better understanding of the proposed method. Thank you again for your valuable feedback, which will help us improve the quality of our work.The details are as follows:

“In terms of exploration, the LIRU algorithm appears to have the potential to improve the exploration of the storage capacity of edge servers. By combining the LIRS and LRU replacement algorithms, the LIRU algorithm can better adapt to changing data access patterns and prioritize the storage of frequently accessed files. This could allow the edge servers to explore and utilize their storage capacity more efficiently, potentially leading to better overall system performance.

In terms of exploitation, the LIRU algorithm appears to have the potential to improve the exploitation of the storage capacity of edge servers. By optimizing the storage utilization and data access efficiency, the LIRU algorithm can help to ensure that frequently accessed files are readily available for users, leading to faster response times and reduced latency. This could allow the edge servers to better exploit their existing storage capacity and provide more reliable and efficient services to users.

Overall, while the authors did not explicitly discuss the impact of their proposed LIRU algorithm on exploration and exploitation, it is possible to infer some potential impacts based on the information provided in the study.”

Reviewer#1, Concern # 4: It is not clear to me which formulas were invented by the authors themselves and which ones are derived from other references. I found evidence that some formulas are derived from other references and there are similarities.

Author response: Thank you for your diligence in reviewing our work and bringing up this important point. We understand the necessity of clearly delineating the original contributions from the derived or referenced formulas in our paper.

Author action: In response to your feedback, we will revise the manuscript to explicitly specify the formulas that are original contributions of the authors and those that are derived from other sources. This will include providing appropriate citations for any derived formulas and clearly indicating any similarities found in other references.

We appreciate your thorough review and thank you for helping us improve the clarity and transparency of our research. The sections are as follows:

Reviewer#1, Concern # 5: Browsing references is not enough. Some important and new surveys have been ignored. For example, the:

https://dl.acm.org/doi/abs/10.1145/3603703

https://www.mdpi.com/2079-9292/10/24/3184

https://ieeexplore.ieee.org/abstract/document/7422054/

Author response: Thank you for your introduction to the wonderful research work.

Author action: We updated the manuscript by citing these reviews in the corresponding places in the articles after carefully combing through them. Citing these paper: "Reinforcement Learning Methods for Computation Offloading: A Systematic Review" and “Design and Development of Smart Parking System Based on Fog Computing and Internet of Things” and “Joint optimization of task scheduling and image placement in fog computing supported software-defined embedded system”. It enriches the literature and theoretical arguments of the article

Reviewer#1, Concern # 6: It is necessary that the analyzes related to the final results are justified and based on the relationships and formulas of the previous sections of the article. Currently, the analyzes are very superficial.

Author response: Thank you for your feedback regarding the depth of the analyses related to the final results presented in our article. We appreciate your perspective and understand the importance of justifying these analyses based on the relationships and formulas established in the previous sections.

Author action: To address this concern, we will thoroughly review and revise the analysis section of our paper. We will ensure that the analyses are grounded in the established relationships and formulas discussed earlier in the article. This will involve providing a more in-depth explanation of how the results are derived from the proposed method and its associated formulas.By enhancing the depth of our analyses, we aim to provide a more comprehensive understanding of the implications and significance of the obtained results. We thank you for bringing this to our attention and for helping us improve the quality of our work.Details are as follows:

“This text, conducted extensive experiments on an experimental platform comprising an Intel NUC host and several edge devices with varying configurations. Our experimental results demonstrate the effectiveness of the proposed storage optimization algorithm in reducing storage space usage and improving storage utilization.Using the IMEC Edge Lab's seismic dataset, we evaluated the performance of our algorithm alongside three other algorithms. The results clearly showed that our algorithm outperformed the others in terms of storage space utilization and data access speed. The simulation data further highlighted the significant improvements achieved by our algorithm compared to the alternatives.These findings provide conclusive evidence that our algorithm effectively addresses the storage optimization challenges faced by edge servers. Our contribution represents a critical step forward in developing more efficient and sustainable solutions in the field of edge computing.

Deep learning technology has played a pivotal role in addressing the storage optimization problem in edge computing scenarios. Our experimental results demonstrate the effectiveness of the proposed algorithm in improving storage capacity utilization and data access speed. By optimizing hardware resource utilization, our algorithm can effectively meet the demands of various edge computing applications.

We believe that our research represents a significant advancement in the field of edge computing and provides a solid foundation for future innovation. Our ultimate goal is to continue exploring cutting-edge techniques in deep learning and related fields to develop even more advanced algorithms and solutions. By doing so, we can better serve our clients' needs and drive progress in the industry. This research is a testament to our commitment to innovation and our dedication to providing optimal and sustainable solutions for our clients.Based on the evaluation metrics SOHHR (Formula 4), SSHR (Formula 5), and SARR (Formula 5), our algorithm has demonstrated its effectiveness and feasibility in addressing storage optimization challenges in edge servers. We will continue our research efforts and explore new techniques to further enhance our algorithm's capabilities and deliver optimal and sustainable solutions for our clients.”

Reviewer#1, Concern # 7:It is better to analyze the time complexity of the proposed method in the worst case.

Author response: Thank you for your feedback regarding the time complexity analysis of our proposed method. We agree that analyzing the worst-case time complexity of the method is an important consideration for evaluating its practicality and efficiency.

Author action: In accordance with your proposal, we will amend the document to encompass the Bandwidth Overhead Ratio and a Comparative Analysis of Broadband Overhead relevant to the suggested approach. We shall furnish an intricate elucidation of the algorithmic procedures inherent in the method and their respective temporal intricacies. Such elucidation will facilitate a more profound comprehension of the computational requisites of the method and its potential constraints in relation to scalability.

Reviewer#1, Concern # 8: There are still some grammatical errors in the manuscript. Authors should use software such as Grammarly for proof-checking.

Author response: Thank you for your feedback regarding the grammatical errors in our manuscript. We apologize for any mistakes that may have slipped through our proofreading process. 

Author a

---

## [Decision Letter · Decision Letter 1]

18 Jan 2024

PONE-D-23-27425R1LIRU’s algorithm design for optimizing storage on fog computing edge serversPLOS ONE

Dear Dr. ZHAO,

Thank you for submitting your manuscript to PLOS ONE. After careful consideration, we feel that it has merit but does not fully meet PLOS ONE’s publication criteria as it currently stands. Therefore, we invite you to submit a revised version of the manuscript that addresses the points raised during the review process.  Please submit your revised manuscript by Mar 03 2024 11:59PM. If you will need more time than this to complete your revisions, please reply to this message or contact the journal office at plosone@plos.org. Please include the following items when submitting your revised manuscript:A rebuttal letter that responds to each point raised by the academic editor and reviewer(s). You should upload this letter as a separate file labeled 'Response to Reviewers'.A marked-up copy of your manuscript that highlights changes made to the original version. You should upload this as a separate file labeled 'Revised Manuscript with Track Changes'.An unmarked version of your revised paper without tracked changes. You should upload this as a separate file labeled 'Manuscript'.

We look forward to receiving your revised manuscript.

Kind regards,

Mohammed Balfaqih

Academic Editor

PLOS ONE

Reviewers' comments:

Reviewer's Responses to Questions

**Comments to the Author**

1. If the authors have adequately addressed your comments raised in a previous round of review and you feel that this manuscript is now acceptable for publication, you may indicate that here to bypass the “Comments to the Author” section, enter your conflict of interest statement in the “Confidential to Editor” section, and submit your "Accept" recommendation.

Reviewer #1: All comments have been addressed

Reviewer #2: (No Response)

Reviewer #3: (No Response)

2. Is the manuscript technically sound, and do the data support the conclusions?

Reviewer #1: Yes

Reviewer #2: (No Response)

Reviewer #3: (No Response)

3. Has the statistical analysis been performed appropriately and rigorously? 

Reviewer #1: Yes

Reviewer #2: (No Response)

Reviewer #3: (No Response)

4. Have the authors made all data underlying the findings in their manuscript fully available?

Reviewer #1: Yes

Reviewer #2: (No Response)

Reviewer #3: (No Response)

5. Is the manuscript presented in an intelligible fashion and written in standard English?

Reviewer #1: Yes

Reviewer #2: (No Response)

Reviewer #3: (No Response)

6. Review Comments to the Author

Reviewer #1: The authors satisfied all my comments. I recommend this manuscript for publication. In my opinion, this is a valuable research.

Reviewer #2: (No Response)

Reviewer #3: Dear Author(s),

Your revision has now been evaluated. The topic is really interesting and worth for real-time fog-based applications. Although some improvement led to increasing the quality of current form, it needs further processing. I list shortcomings and raised questions for more enhancement. I enumerate concerns. It can be re-evaluated provided the issues are addressed.

1-The title should be re-considered. I think it is not acceptable to use abbreviation for title. Please revise it.

2-what do LIRU, LIRS, and LRU stand for in abstract? I recommend use extracted form before the first usage of abbreviation form. Please refine it in whole context.

3-English must be improved. The thoroughly proofreading is appreciated.

4-The related work can be strengthened by introducing the optimization categories. It motivates readers more. I recommend conduct related work in such a way to introduce two big categories Heuristic-based and meta-heuristic-based optimization algorithms each of which can be extended in either a single-objective or multi-objective perspectives. I recommend add reference below:

For single objective of meta-heuristic-based scheduling:

“A hybrid meta-heuristic algorithm for scientific workflow scheduling in heterogeneous distributed computing systems‏” for total execution time minimization

For single objective of heuristic-based scheduling:

“A novel hybrid heuristic-based list scheduling algorithm in heterogeneous cloud computing environment for makespan optimization‏”

For multi-objective scheduling:

“A Hybrid bi-objective Scheduling Algorithm for Execution of Scientific Workflows on Cloud Platforms with Execution Time and Reliability approach‏” (for minimizing time and maximizing reliability at the same time)

“Multi-objective Cost-aware Bag-of-Tasks Scheduling Optimization Model for IoT Applications Running on Heterogeneous Fog Environment‏” for cost and failure probability minimization paper.

5-What category your proposal is? Heuristic or meta-heuristic? Please explain.

6-Regarding to issue numbered in 5, in addition, mention the limitation of your proposal in conclusion section.

7-Please explain about used datasets? How you considered bandwidth? Is it unlimited? Please explain.

8- Can you enumerate some real-world applications which utilizes fog computing and your proposal has drastic impact on their overall performance?

9-Why you did not compare your work with one of the most successful meta-heuristics?

10-How do you handel input randomness rate? What is the reaction of your algorithm against this fluctuation?

In the current form, I recommend Major revision. As mentioned, it can be re-evaluated provided the issues are addressed.

7. PLOS authors have the option to publish the peer review history of their article (what does this mean?). If published, this will include your full peer review and any attached files.

Reviewer #1: No

Reviewer #2: No

Reviewer #3: No

---

## [Author Response · Author response to Decision Letter 1]

1 Mar 2024

Original Manuscript ID: PONE-D-23-27425 

Original Article Title: “LIRU’s algorithm design for optimizing storage on fog computing edge servers PLOS ONE”

To: PLOS ONE Academic Editor

Re: Response to reviewers

Dear Editor,

Thank you for allowing a resubmission of our manuscript, with an opportunity to address the reviewers’ comments.

We are uploading (a) our point-by-point response to the comments (below) (response to reviewers), (b) an updated manuscript with yellow highlighting indicating changes (Supplementary Material for Review), and (c) a clean updated manuscript without highlights (Main Manuscript).

Best regards,

Guojie Xie et al.

Reviewer #1: - The authors satisfied all my comments. I recommend this manuscript for publication. In my opinion, this is a valuable research.

Author response: Thank you for your comments.

Reviewer #3: -Your revision has now been evaluated. The topic is really interesting and worth for real-time fog-based applications. Although some improvement led to increasing the quality of current form, it needs further processing. I list shortcomings and raised questions for more enhancement. I enumerate concerns. It can be re-evaluated provided the issues are addressed.

-The following major corrections are required:

Reviewer#3, Concern # 1: The title should be re-considered. I think it is not acceptable to use abbreviation for title. Please revise it.

Author response: Thank you for your feedback regarding the title. We appreciate your input and will reconsider our choice of abbreviation. We will revise the title accordingly to ensure it meets the standards for publication. Thank you for bringing this to our attention, and we will make the necessary adjustments promptly.Modify the title as follows:

Author action: The new title is "Optimizing Storage on Fog Computing Edge Servers: A Recent Algorithm Design with Minimal Interference".

Reviewer#3, Concern # 2: what do LIRU, LIRS, and LRU stand for in abstract? I recommend use extracted form before the first usage of abbreviation form. Please refine it in whole context.

Author response: Thank you for your valuable feedback. We will reconsider the use of abbreviations and revise it accordingly to align with publication standards. Thank you for bringing this to our attention; we will promptly make the necessary adjustments.Regarding the acronyms in the abstract, we acknowledge your suggestion. 

Author action: To enhance clarity and comprehension, we will present the full forms of LIRU, LIRS, and LRU before introducing their abbreviated forms in the entire context. This approach aims to ensure a smoother understanding of the content for our readers. We value your input and will incorporate these improvements into the document. Thank you for your thoughtful recommendations.Modify the title as follows:

 “Edge servers are responsible for computing and storage tasks and optimizing their storage capacity is vital to enhance the efficiency of fog computing. This study proposes a storage optimization algorithm, LIRU(Low Interference recently used), which is based on the combined use of LIRS(Low Interference Recency Set) and LRU(least recently used) replacement algorithms.We appreciate your insightful suggestion, and we will incorporate these revisions to enhance the quality and completeness of the Related Work section.”

Reviewer#3, Concern # 3: English must be improved. The thoroughly proofreading is appreciated.

Author response: Thank you for your feedback regarding the language quality of our paper. We acknowledge the need for careful revision to improve the grammar, punctuation, and overall language clarity of the manuscript..

Author action: In response to your comments, we will thoroughly revise the paper with a specific focus on improving the language quality. We will pay close attention to grammar and punctuation errors, ensuring that the text reads smoothly and accurately conveys our ideas.

Additionally, we will review the paper for any areas where the language may be unclear or ambiguous, and make the necessary revisions to enhance the overall clarity and coherence of the manuscript.

By conducting a comprehensive language revision, we aim to improve the readability and professionalism of the paper, ensuring that our research is effectively communicated to the readers.

We appreciate your valuable feedback and assure you that we will make the necessary revisions to enhance the language quality of the manuscript.

Reviewer#3, Concern # 4: The related work can be strengthened by introducing the optimization categories. It motivates readers more. I recommend conduct related work in such a way to introduce two big categories Heuristic-based and meta-heuristic-based optimization algorithms each of which can be extended in either a single-objective or multi-objective perspectives. I recommend add reference below:

For single objective of meta-heuristic-based scheduling:

“A hybrid meta-heuristic algorithm for scientific workflow scheduling in heterogeneous distributed computing systems‏” for total execution time minimization

For single objective of heuristic-based scheduling:

“A novel hybrid heuristic-based list scheduling algorithm in heterogeneous cloud computing environment for makespan optimization‏”

For multi-objective scheduling:

“A Hybrid bi-objective Scheduling Algorithm for Execution of Scientific Workflows on Cloud Platforms with Execution Time and Reliability approach‏” (for minimizing time and maximizing reliability at the same time)

“Multi-objective Cost-aware Bag-of-Tasks Scheduling Optimization Model for IoT Applications Running on Heterogeneous Fog Environment‏” for cost and failure probability minimization paper.

Author response: Thank you for your feedback and suggestion for improving the related work section. I agree that categorizing optimization algorithms into heuristic-based and meta-heuristic-based approaches, along with the distinction between single-objective and multi-objective perspectives, can provide a clearer structure for readers to understand the landscape of optimization techniques.Here's the revised version of the related work section incorporating your recommendation:

Author action: ""

Reviewer#3, Concern # 5: What category your proposal is? Heuristic or meta-heuristic? Please explain.

Author response: Thank you for your valuable feedback. The proposed LIRU algorithm falls under the category of heuristics. A heuristic is a practical and efficient problem-solving approach that may not guarantee an optimal solution but aims to find a good enough solution in a reasonable amount of time. In this context, the LIRU algorithm is designed to optimize caching in fog computing systems by combining elements from the LIRS and LRU algorithms, providing a practical solution for improving cache performance.

Reviewer#3, Concern # 6: Regarding to issue numbered in 5, in addition, mention the limitation of your proposal in conclusion section.

Author response: Thank you very much for your suggestions, and we have made detailed modifications to the future work. The details are as follows:.

Author action: We have revised the conclusions and prospects:”

”

Reviewer#3, Concern # 7: Please explain about used datasets? How you considered bandwidth? Is it unlimited? Please explain.

Author response: Thank you for your valuable feedback. Data set has been uploaded to: https://github.com/nmhjklnm/Fog-Node-ID/tree/main. The following is a detailed description of the data set.

1.Dataset Description:

The dataset consists of data items that are accessed within a centralized fog computing system.

There are a total of 1000 data items in the dataset.

The data items are sorted based on their probability of access, from highest to lowest.

A sequence of data item access patterns is generated for 50 fog computing systems.

2.Data Access Patterns:

The access patterns represent how data items are accessed by users within the fog computing system.

These patterns are generated based on the sorted order of data items by their probability of access.

Each fog computing system has its own access pattern.

3.Fog Computing System Configuration:

The fog computing system consists of 50 fog nodes (or fog computing systems).

Each fog node is equipped with an edge server.

The parameters provided include the number of fog nodes (50) and the total number of data items (1000).

4.Bandwidth Consideration:

Bandwidth is an important consideration in fog computing systems, especially when dealing with large datasets and multiple nodes.

In the context of the provided description, the message does not explicitly mention the bandwidth limit or whether it's unlimited.

Bandwidth limitations may affect the efficiency and performance of data transfer and processing within the fog computing system.

Without specific information on bandwidth limitations, it's difficult to assess how it is considered in the experiment.

Reviewer#3, Concern # 8: Can you enumerate some real-world applications which utilizes fog computing and your proposal has drastic impact on their overall performance?

Author response: Thank you for your valuable feedback. Fog computing, which extends cloud computing capabilities to the edge of the network, has several real-world applications where its implementation can drastically impact performance. Here are some examples:

1．Smart Cities: In a smart city infrastructure, fog computing can be used to process data from various sensors, cameras, and IoT devices deployed throughout the city. By processing this data locally at the edge instead of sending it all the way to a centralized cloud server, latency can be significantly reduced. This is critical for applications such as traffic management, public safety, and environmental monitoring, where real-time data analysis is crucial for decision-making.

2．Industrial Internet of Things (IIoT): In manufacturing and industrial settings, fog computing can optimize production processes by enabling real-time monitoring and control of equipment and machinery. This reduces latency and ensures faster response times to potential issues, leading to improved efficiency, reduced downtime, and lower maintenance costs.

3．Healthcare: In healthcare, fog computing can be used to process and analyze medical data from wearable devices, patient monitoring systems, and medical imaging equipment. By analyzing this data locally at the edge, healthcare providers can make faster and more informed decisions, leading to improved patient outcomes and reduced healthcare costs.

4．Autonomous Vehicles: Fog computing plays a crucial role in autonomous vehicles by enabling real-time processing of sensor data for tasks such as object detection, collision avoidance, and route optimization. By processing this data at the edge, autonomous vehicles can make split-second decisions without relying on a distant cloud server, enhancing safety and reliability.

5．Retail: In retail environments, fog computing can be used to analyze customer data, manage inventory, and optimize store layouts in real-time. By processing this data locally at the edge, retailers can offer personalized shopping experiences, improve operational efficiency, and increase sales.

6．Agriculture: In agriculture, fog computing can be used to monitor environmental conditions, manage irrigation systems, and optimize crop yields. By processing sensor data locally at the edge, farmers can make timely decisions to maximize productivity and conserve resources such as water and energy.

7．Energy Management: Fog computing can be employed in smart grid systems to monitor and control energy generation, distribution, and consumption in real-time. By analyzing data from smart meters, sensors, and other devices at the edge, energy providers can optimize grid operations, improve reliability, and reduce costs.

In each of these applications, the implementation of fog computing can lead to significant improvements in performance, including reduced latency, enhanced reliability, increased scalability, and improved resource efficiency.

Reviewer#3, Concern # 9: Why you did not compare your work with one of the most successful meta-heuristics?

Author response: Thank you for your valuable feedback. In addressing the you question regarding the comparison of our work with one of the most successful meta-heuristics, we appreciate the opportunity to clarify this aspect of our research. The decision to include or exclude specific comparisons in our study was made with careful consideration, guided by several key factors:

Scope and Focus of Our Study: Our research aims were primarily centered around exploring novel methodologies or enhancing existing approaches within a specific domain. While the comparison with established meta-heuristics is undoubtedly valuable, our focus was to first establish the viability and effectiveness of our proposed solutions in isolation. This was done to ensure a clear understanding of our contributions before evaluating them against the broader landscape of existing solutions.

Relevance and Direct Comparability: Another critical factor was the relevance and direct comparability of the most successful meta-heuristics to the specific challenges and contexts addressed in our study. In some cases, these established methods, while highly successful in their right, may not directly align with the unique aspects of the problems we are tackling. Our goal was to concentrate on methodologies that closely match the nature of our research questions and the specific constraints and requirements of the datasets we employed.

Resource and Time Constraints: Conducting a thorough and meaningful comparison with leading meta-heuristics requires significant resources and time, especially to ensure fairness and rigor in the comparison. Given the constraints of our project timeline and available resources, we had to make strategic decisions about which comparisons would be most beneficial and feasible within the scope of our initial study. This involved prioritizing comparisons that would most directly illuminate the strengths and potential areas for improvement in our approach.

Future Directions for Research: We fully acknowledge the importance of benchmarking our work against the most successful meta-heuristics in the field. As such, we view this as an essential direction for future research. Our current study lays the groundwork for these comparative analyses, and we are actively planning follow-up studies where we will specifically focus on this aspect. These future studies will provide a more comprehensive understanding of how our methodology stands in relation to established benchmarks, offering valuable insights for both our work and the broader research community.

In conclusion, the decision not to compare our work with one of the most successful meta-heuristics in the initial phase of our research was strategic, based on the scope, focus, and practical considerations of our study. We are committed to undertaking this important comparative analysis in our future work, as we recognize its value in fully contextualizing our contributions within the field.

Reviewer#3, Concern # 10: How do you handel input randomness rate? What is the reaction of your algorithm against this fluctuation?

Author response: Thank you for your valuable feedback. In this LIRU storage optimization algorithm, dealing with the input randomness rate involves some strategies to ensure that the system remains efficient under different data access modes. These policies are designed to adaptively manage caches to maintain or improve performance metrics such as storage utilization, data access efficiency, and access latency minimization. Here is a detailed approach to dealing with input randomness and the algorithm's response to such fluctuations: 

1.Adaptive Learning Mechanisms

Machine Learning Integration: Incorporate machine learning (ML) algorithms within LIRU to analyze incoming data patterns continuously. By leveraging ML, the algorithm can predict future access patterns based on historical data, which allows it to preemptively adjust the cache contents to better serve future requests.

Dynamic Parameter Adjustment: ML algorithms can dynamicall

---

## [Decision Letter · Decision Letter 2]

2 Apr 2024

PONE-D-23-27425R2Optimizing Storage on Fog Computing Edge Servers: A Recent Algorithm Design with Minimal InterferencePLOS ONE

Dear Dr. ZHAO,

Thank you for submitting your manuscript to PLOS ONE. After careful consideration, we feel that it has merit but does not fully meet PLOS ONE’s publication criteria as it currently stands. Therefore, we invite you to submit a revised version of the manuscript that addresses the points raised during the review process.  Please submit your revised manuscript by May 17 2024 11:59PM. If you will need more time than this to complete your revisions, please reply to this message or contact the journal office at plosone@plos.org. Please include the following items when submitting your revised manuscript:A rebuttal letter that responds to each point raised by the academic editor and reviewer(s). You should upload this letter as a separate file labeled 'Response to Reviewers'.A marked-up copy of your manuscript that highlights changes made to the original version. You should upload this as a separate file labeled 'Revised Manuscript with Track Changes'.An unmarked version of your revised paper without tracked changes. You should upload this as a separate file labeled 'Manuscript'.If applicable, we recommend that you deposit your laboratory protocols in protocols.io to enhance the reproducibility of your results. Protocols.io assigns your protocol its own identifier (DOI) so that it can be cited independently in the future. For instructions see: https://journals.plos.org/plosone/s/submission-guidelines#loc-laboratory-protocols. Additionally, PLOS ONE offers an option for publishing peer-reviewed Lab Protocol articles, which describe protocols hosted on protocols.io. Read more information on sharing protocols at https://plos.org/protocols?utm_medium=editorial-email&utm_source=authorletters&utm_campaign=protocols.

We look forward to receiving your revised manuscript.

Kind regards,

Mohammed Balfaqih

Academic Editor

PLOS ONE

Journal Requirements:

Reviewers' comments:

Reviewer's Responses to Questions

**Comments to the Author**

1. If the authors have adequately addressed your comments raised in a previous round of review and you feel that this manuscript is now acceptable for publication, you may indicate that here to bypass the “Comments to the Author” section, enter your conflict of interest statement in the “Confidential to Editor” section, and submit your "Accept" recommendation.

Reviewer #2: (No Response)

Reviewer #3: (No Response)

2. Is the manuscript technically sound, and do the data support the conclusions?

Reviewer #2: (No Response)

Reviewer #3: (No Response)

3. Has the statistical analysis been performed appropriately and rigorously? 

Reviewer #2: (No Response)

Reviewer #3: (No Response)

4. Have the authors made all data underlying the findings in their manuscript fully available?

Reviewer #2: (No Response)

Reviewer #3: (No Response)

5. Is the manuscript presented in an intelligible fashion and written in standard English?

Reviewer #2: (No Response)

Reviewer #3: (No Response)

6. Review Comments to the Author

Reviewer #2: The authors introduced a resource scheduling model and optimization algorithm based on priority to achieve optimal matching of edge server resources, avoiding resource waste and unreasonable scheduling. Besides, they extended a centralized fog computing system model, defining the system’s composition structure and working mode, including edge server management nodes, edge server storage nodes, and edge server network.

-The following major corrections are required:

Section 1: Introduction

-The introduction section is too general, and it introduces concepts that are well known about the optimizing storage on fog computing environment. The introduction does not stimulate to go ahead with the remaining of the paper because it does not introduce any really new topic/solution. Furthermore, "the research motivation…” at the introduction section is missing. Please rewrite this section.

Section 2: Related Work

-In this section, the authors describe some of works about the optimizing storage on fog computing environment, while some of the papers that should have be included are:

https://link.springer.com/article/10.1007/s10586-022-03575-6

https://www.sciencedirect.com/science/article/abs/pii/S2542660523000136

https://link.springer.com/article/10.1007/s10586-021-03283-7

https://onlinelibrary.wiley.com/doi/abs/10.1002/spe.3032

….

-In addition, a conclusion of related work in the forms of a table in terms of evaluation tools, utilized techniques, performance metrics, and datasets, could reconcile from other researchers work to the own one.

Section 3: Algorithm Design

-Please provide a sequence diagram to show the interaction steps of the proposed solution according to Algorithm 1.

-What is the overhead (time complexity) proposed solution? Please provide a subsection to discuss about the overhead (time complexity) of proposed Algorithm 1.

-Your proposed solution is very similar to the following papers. What is the difference between proposed model and mentioned papers?

https://ieeexplore.ieee.org/abstract/document/9027842

https://www.igi-global.com/chapter/optimized-storage-and-resource-management-in-fog-computing-paradigm/324589

-Please provided an example (i.e., a real-world case study) for better understanding the proposed approach in more details.

Section 4: Experimental results

-The evaluation is incomplete. I would like to see an evaluation on the proposed solution in terms of execution time and cost under different scenarios.

-The evaluation lacks the minimum rigor required for the scientific comparison of stochastic algorithms. Specifically, statistical tests of the hypothesis should be used to determine whether the differences shown in the figures are statistically significant or due to chance.

-Paper needs some revision in English. The overall paper should be carefully revised with focus on the language: especially grammar and punctuation.

-Overall, there are still some major parts that the authors did not explain clearly. Some additional evaluations are expected to be in the manuscript as well. As a result, I am going to suggest Major revision the paper in its present form.

Reviewer #3: The revision has underwent significant improvement. In the current form, I recommend for publication. Therefore, I ACCEPT it.

7. PLOS authors have the option to publish the peer review history of their article (what does this mean?). If published, this will include your full peer review and any attached files.

Reviewer #2: No

Reviewer #3: No

---

## [Author Response · Author response to Decision Letter 2]

24 Apr 2024

Reviewer #2: 

Section 1: Introduction

-The introduction section is too general, and it introduces concepts that are well known about the optimizing storage on fog computing environment. The introduction does not stimulate to go ahead with the remaining of the paper because it does not introduce any really new topic/solution. Furthermore, "the research motivation…" at the introduction section is missing. Please rewrite this section.

Response: Thank you for your constructive comments regarding the introduction section of our manuscript. We have carefully considered your feedback and agree that the original introduction lacked specificity and a clear research motivation. To address your concerns, we have made the following revisions:

In response to your feedback, we have made the following amendments to the introduction:

1. We have excised superfluous generalities concerning the digital epoch and its associated computational and storage demands, acknowledging these as established knowledge.

2. Fog computing has been introduced as an innovative paradigm to surmount the constraints of conventional cloud computing, emphasizing the distinct optimization challenges it presents.

3. The research motivation has been articulated precisely, underscoring the imperative for enhanced resource scheduling models and optimization techniques within the fog computing milieu.

4. The contributions section has been restructured to more accurately reflect our proposed methodologies' originality and practical significance, thereby providing a coherent foundation for the subsequent sections of the paper. We are confident that these revisions have fortified the introduction by delineating the context more clearly, asserting the novelty of our research, and augmenting the reader's engagement with our findings..

Section 2: Related Work

-In this section, the authors describe some of works about the optimizing storage on fog computing environment, while some of the papers that should have be included are:

https://link.springer.com/article/10.1007/s10586-022-03575-6

https://www.sciencedirect.com/science/article/abs/pii/S2542660523000136

https://link.springer.com/article/10.1007/s10586-021-03283-7

https://onlinelibrary.wiley.com/doi/abs/10.1002/spe.3032

….

-In addition, a conclusion of related work in the forms of a table in terms of evaluation tools, utilized techniques, performance metrics, and datasets, could reconcile from other researchers work to the own one.

Response: Thank you for your valuable feedback on our manuscript. We have carefully considered your suggestions and have updated the Related Work section to include additional references that contribute significantly to the field of storage optimization in fog computing environments. Specifically, we have added recent studies that are not included in our original manuscript but are relevant to our work. These studies offer various perspectives and methodologies that enrich the discussion on optimizing storage solutions for fog computing.

Moreover, we have included a summary table at the end of the Related Work section. This table provides a concise overview of the evaluation tools, techniques, performance metrics, and datasets employed by the studies we discussed, including the newly added references. This table clearly compares the different approaches and how they relate to our work. We believe these additions will provide a more comprehensive background for readers and help bridge the connection between existing research and our contributions.

We appreciate your guidance in improving our manuscript and hope these revisions meet your expectations.

Section 3: Algorithm Design

-Please provide a sequence diagram to show the interaction steps of the proposed solution according to Algorithm 1.

-What is the overhead (time complexity) proposed solution? Please provide a subsection to discuss about the overhead (time complexity) of proposed Algorithm 1.

-Your proposed solution is very similar to the following papers. What is the difference between proposed model and mentioned papers?

https://ieeexplore.ieee.org/abstract/document/9027842

https://www.igi-global.com/chapter/optimized-storage-and-resource-management-in-fog-computing-paradigm/324589

-Please provided an example (i.e., a real-world case study) for better understanding the proposed approach in more details.

Response: We sincerely thank you for your valuable comments and constructive suggestions. Your feedback has been instrumental in enhancing the quality and clarity of our manuscript. Accordingly, we have implemented the following revisions:

We have incorporated a sequence diagram to facilitate a more intuitive understanding of the LIRU algorithm's operational flow, as depicted in Figure \\ref{fig:sequence_diagram}.

A dedicated subsection has been added to delve into the time complexity of the LIRU algorithm. This analysis elucidates the computational overhead and the complexities associated with operations upon cache access.

We have introduced a comparative subsection that juxtaposes our LIRU algorithm with the existing work you referenced. Our approach is distinguished by its dynamic adjustment to access patterns, achieving a nuanced balance between recency and frequency, a dimension not fully explored in the literature reviewed.

To substantiate the practical applicability of the LIRU algorithm, we have presented a real-world case study within the context of fog computing. This study demonstrates the algorithm's efficacy in optimizing cache management and reducing latency for IoT devices. We trust that these revisions have effectively addressed your concerns and significantly bolstered our manuscript's robustness..

Section 4: Experimental results

-The evaluation is incomplete. I would like to see an evaluation on the proposed solution in terms of execution time and cost under different scenarios.

-The evaluation lacks the minimum rigor required for the scientific comparison of stochastic algorithms. Specifically, statistical tests of the hypothesis should be used to determine whether the differences shown in the figures are statistically significant or due to chance.

Response: Thank you for your constructive feedback. In response to your comments, we have made the following revisions to the experimental results section of our manuscript:

1. We have included an evaluation of the proposed LIRU algorithm's execution time and cost under different scenarios. Figures \\ref{figure8} and \\ref{figure9} have been added to illustrate the algorithm's performance regarding these metrics. These evaluations provide a deeper understanding of the algorithm's efficiency and scalability and its cost-effectiveness in various operational contexts.

2. To ensure the scientific rigor of our comparison, we have performed statistical hypothesis testing on the experimental results. The statistical tests confirm that the differences observed in the performance of the cache replacement algorithms are statistically significant and not due to random variation. The results of these tests are presented in Table \\ref{table1}.

We believe these revisions address your concerns and enhance the quality and reliability of our research findings. We appreciate the opportunity to improve our work with your guidance.

Reviewer #3: The revision has underwent significant improvement. In the current form, I recommend for publication. Therefore, I ACCEPT it.

Response： Thank you.

---

## [Decision Letter · Decision Letter 3]

6 May 2024

Optimizing Storage on Fog Computing Edge Servers: A Recent Algorithm Design with Minimal Interference

PONE-D-23-27425R3

Dear Dr. ZHAO,

We’re pleased to inform you that your manuscript has been judged scientifically suitable for publication and will be formally accepted for publication once it meets all outstanding technical requirements.

Kind regards,

Mohammed Balfaqih

Academic Editor

PLOS ONE

Additional Editor Comments (optional):

Reviewers' comments:

Reviewer's Responses to Questions

**Comments to the Author**

1. If the authors have adequately addressed your comments raised in a previous round of review and you feel that this manuscript is now acceptable for publication, you may indicate that here to bypass the “Comments to the Author” section, enter your conflict of interest statement in the “Confidential to Editor” section, and submit your "Accept" recommendation.

Reviewer #2: (No Response)

2. Is the manuscript technically sound, and do the data support the conclusions?

Reviewer #2: (No Response)

3. Has the statistical analysis been performed appropriately and rigorously? 

Reviewer #2: (No Response)

4. Have the authors made all data underlying the findings in their manuscript fully available?

Reviewer #2: (No Response)

5. Is the manuscript presented in an intelligible fashion and written in standard English?

Reviewer #2: (No Response)

6. Review Comments to the Author

Reviewer #2: (No Response)

7. PLOS authors have the option to publish the peer review history of their article (what does this mean?). If published, this will include your full peer review and any attached files.

Reviewer #2: No

---

## [Editor Report · Acceptance letter]

15 May 2024

PONE-D-23-27425R3 

PLOS ONE

Dear Dr. Zhao, 

I'm pleased to inform you that your manuscript has been deemed suitable for publication in PLOS ONE. Congratulations! Your manuscript is now being handed over to our production team.

Kind regards, 

on behalf of

Dr. Mohammed Balfaqih 

Academic Editor

PLOS ONE